# Discovery, characterization, and comparative analysis of new UGT72 and UGT84 family glycosyltransferases
Tuo Li[1,4], Annika J. E. Borg[1,4], Leo Krammer [2], Hansjörg Weber[2], Rolf Breinbauer [2] & Bernd Nidetzky [1,3] ✉

Glycosylated derivatives of natural product polyphenols display a spectrum of biological activities, rendering them critical for both nutritional and pharmacological applications. Their enzymatic synthesis by glycosyltransferases is frequently constrained by the limited repertoire of characterized enzyme-catalyzed transformations. Here, we explore the glycosylation capabilities and substrate preferences of newly identified plant uridine diphosphate (UDP)-dependent glycosyltransferases (UGTs) within the UGT72 and UGT84 families, with particular focus on natural polyphenol glycosylation from UDP-glucose. Four UGTs are classified according to their phylogenetic relationships and reaction products, identifying them as biocatalysts for either glucoside (UGT72 enzymes) or glucose ester (UGT84 members) formation from selected phenylpropanoid compounds. Detailed kinetic evaluations expose the unique attributes of these enzymes, including their specific activities and regio-selectivities towards diverse polyphenolic substrates, with product characterizations validating the capacity of UGT84 family members to perform di-*O*-glycosylation on flavones. Sequence analysis coupled with structural predictions through AlphaFold reveal an unexpected absence of a conserved threonine residue across all four enzymes, a trait previously linked to pentosyltransferases. This comparative analysis broadens the understood substrate specificity range for UGT72 and UGT84 enzymes, enhancing our understanding of their utility in the production of natural phenolic glycosides. The findings from this in-depth characterization provide valuable insights into the functional versatility of UGT-mediated reactions.

Plants contain a broad variety of natural phenolics, which serve as important compounds in food and pharmaceutical industry due to their diverse biological activities[1–3]. Glycosylation presents an effective method for mitigating the challenges posed by low solubility, poor stability, and limited bioavailability of phenolic compounds in their applications[4–6]. Uridine diphosphate (UDP)-dependent glycosyltransferases (UGTs) play a pivotal role in the natural glycosylation and biosynthesis of various phenolic glycosides by transferring UDP-sugars to acceptor substrates[7–10]. The lack of well-characterized UGT-catalyzed transformations limits the practical usage of these enzymes, which presents the need for the discovery of new catalytic activities[11–13].

Plant UGTs typically possess a 44-amino acid consensus sequence known as plant secondary product glycosyltransferase (PSPG) motif, which serves as a useful tool in the bioinformatic search of putative UGT genes[14–16]. Based on sequence analysis, numerous UGTs have been classified into different families and larger groups that include several distinct UGT families[17]. UGTs within the same family display an amino acid sequence identity of greater than 40%, and are assigned with a characteristic numerical designation (e.g., UGT72, UGT84)[18].

Among the 18 major phylogenetic groups of plant UGTs (labeled from A to R based on sequence homology), group E UGTs, which consist of UGT71, UGT72, and UGT88 families, have been shown to produce glucosides of phenolic compounds[17]. As their predicted natural function, UGT72 enzymes play an important role in the biosynthesis of plant lignin and are responsible for the glycosylation of monolignols, encompassing *p*-coumaryl, coniferyl, and sinapyl alcohols, along with their corresponding

[1]Institute of Biotechnology and Biochemical Engineering, Graz University of Technology, NAWI Graz, Petersgasse 12/1, 8010 Graz, Austria. [2]Institute of Organic Chemistry, Graz University of Technology, NAWI Graz, Stremayrgasse 9, 8010 Graz, Austria. [3]Austrian Centre of Industrial Biotechnology (acib), Krenngasse 37, 8010 Graz, Austria. [4]These authors contributed equally: Tuo Li, Annika J. E. Borg. ✉e-mail: bernd.nidetzky@tugraz.at

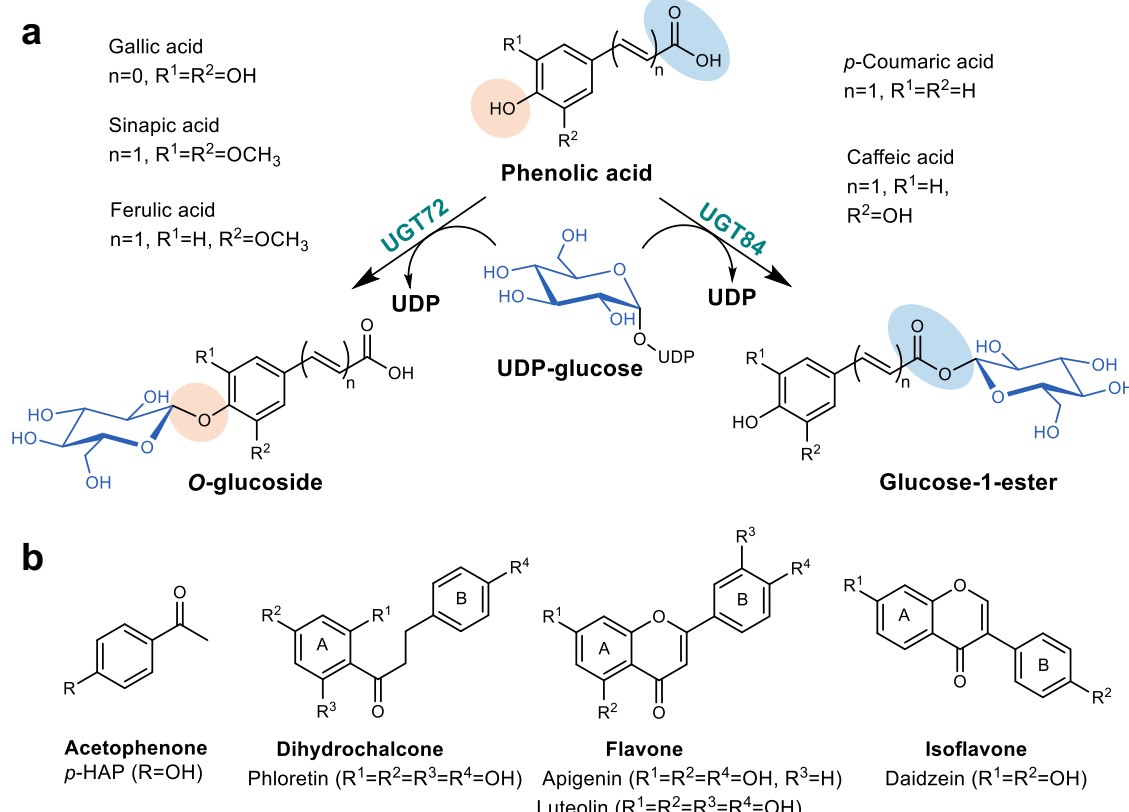

**Fig. 1 | Reaction schemes and potential phenolic substrates for UGT72 and UGT84 candidates. a** Reaction scheme from a phenolic acid to a corresponding *O*-glucoside or glucose-1-ester via UDP-glucose, catalyzed by UGT72 or UGT84 enzymes, respectively. The glycosylation sites are highlighted with orange and blue circles. **b** Backbone structures of selected phenolic substrates (acetophenone, dihydrochalcone, flavone, isoflavone).

aldehydes and acids[19–23]. Similar to most other UGTs, UGT72 enzymes form a glycosidic linkage (e.g., C−O−C, Fig. 1a) from the phenolic hydroxyl group of the aglycone to the anomeric carbon of the sugar moiety[24,25]. Conversely, exclusive glucose ester-forming activities toward phenolic acids are exhibited by group L UGTs, which include UGT74, UGT75, and UGT84 family enzymes[26]. UGT84 enzymes transfer a sugar molecule into the carboxylic acid moiety of sinapic acid, gallic acid, and other phenolic acids, generating ester linkages (C(=O)−O−C, Fig. 1a)[27–32].

Members from UGT72 and UGT84 families often manifest *O*-glycosylation activities toward polyphenolic acceptor substrates (e.g., flavonoids) producing high-value phenolic *O*-glycosides[26,33–37]. Despite the widespread individual characterization of UGT72 and UGT84 enzymes concerning various phenolic substrates, a paucity of comparative studies between these two enzyme classes persists. Furthermore, previous investigations have often lacked a comprehensive kinetic analysis of reaction components, presenting challenges in the application of these UGTs for the biosynthesis of natural phenolic glycosides. Thus, a nuanced understanding of the biochemical attributes and potential utilizations of UGT72 and UGT84 enzymes necessitates further exploration and in-depth investigation of their glycosylation activities.

Here, previously unknown plant UGTs were discovered and thoroughly characterized. The enzymes were classified into UGT72 and UGT84 families based on their phylogenetic relationships and product patterns (glucoside vs glucose ester) obtained from representative phenylpropanoid substrates. A comprehensive analysis of the specific activities and regioselectivities conducted on selected polyphenolic substrate classes (Fig. 1b) revealed distinct family-dependent characteristics of the enzymes. Preparative-scale reactions with product isolations confirmed the di-*O*-glycosylation potential of UGT84 enzymes on flavones. The AlphaFold-predicted 3D-structures of the discovered UGTs highlight the absence of a

conserved threonine for the stabilization of donor glucose C6-OH, a phenomenon previously linked to pentosyltransferases[38]. This research contributes to the advancement of our comprehension of the natural functions, substrate promiscuity, and interconnections associated with UGT72 and UGT84 enzymes.

## Results and discussion
### Discovery and phylogenetic analysis of putative UGT72 and UGT84 enzymes

For initiating the search of putative UGT candidates, well-characterized *Arabidopsis thaliana* enzymes UGT72E3[19] and UGT84A2[39] were chosen as representative members from the UGT72 and UGT84 families, respectively. Using these enzymes as benchmarks, a BLAST search was initiated with a stringent criterion set to identify homologous sequences with an amino acid identity exceeding 40%. This search yielded potential UGT72 and UGT84 family members from a diverse array of plant species, six of which were selected for further investigation (Supplementary Table 1). The herein named putative UGTs UGT84A49 (accession: Q2V6K1), UGT84A119 (accession: A0A2N9FYZ7) and J045 (accession: A0A078J045) shared >50% sequence identity with UGT84A2, while UGT72D1 (accession: Q9ZU72), UGT72D7 (accession: A0A067GVI4) and KZ95 (accession: A0A2U1KZ95) were closely related to UGT72E3 (>40% sequence identity, Supplementary Table 2). Among these genes, only the one corresponding to UGT72D1 had previously been linked to glycosyltransferase activity; however, the enzyme had not been subjected to a detailed analysis concerning substrate specificity and regioselectivity[25]. Despite the high sequence homology between the candidate UGTs and their respective *Arabidopsis* templates, inter-family sequence identities were notably lower, falling beneath the 30% threshold (Supplementary Table 2). Subsequent multiple sequence alignment revealed the conserved C-terminal PSPG motif for the sugar donor binding

**Fig. 2 | Partial sequence alignment and phylogenetic analysis of the putative UGTs together with characterized plant UGTs. a** Partial multiple sequence alignment of the PSPG boxes of putative UGT72 enzymes (UGT72D1, UGT72D7, KZ95), UGT84 enzymes (UGT84A119, UGT84A49, J045) and templates (UGT72E3, UGT84A2).
**b** Phylogenetic analysis of the six putative UGTs with functionally well-characterized plant UGTs. Protein sequences were aligned with ClustalW, and a neighbor-joining tree was constructed by using MEGA11 software (bootstrap-value: 1000). UGTs from 72 and 84 families are highlighted with yellow and red background, respectively. The selected UGT candidates are marked with orange (putative UGT72 enzymes) and blue (putative UGT84 enzymes) rectangles. UGTs from group E (UGT71, UGT72, UGT88) and group L (UGT74, UGT75, UGT84) are indicated by orange and red brackets, respectively.

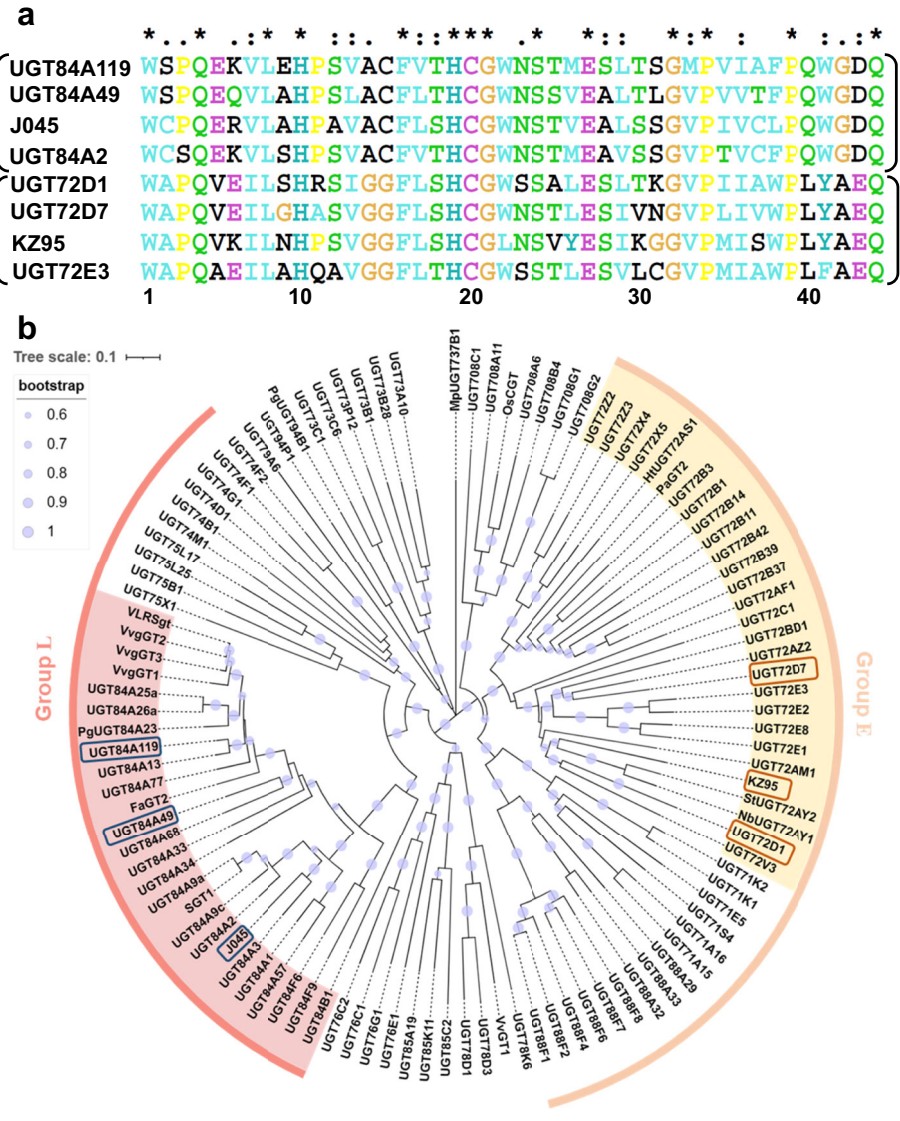

(Fig. 2a)[14,40,41], confirming the chosen candidates as plant UGTs. Remarkably, distinct conserved family-dependent differences were discovered in the following positions of the PSPG-box: Cys/Ser in position 2 for UGT84 members in comparison to Ala in UGT72, Glu in position 5 for UGT84 enzymes in respect to Val/Ala in UGT72 family, Cys in position 15 of UGT84 enzymes versus Gly in UGT72, and Gln in the position 40 of UGT84 members in comparison to Leu in UGT72 (Fig. 2a). The presence of a conserved Gln in the last position 44 of all the enzymes indicates the preference for UDP-glucose (UDP-Glc) as a donor substrate (Fig. 2a)[42].

Phylogenetic analysis of the putative UGTs together with characterized plant UGTs corroborated the taxonomic assignment of UGT72D1, UGT72D7, and KZ95 within the UGT72 family, belonging to the larger phylogenetic group E (Fig. 2b, Supplementary Table 3). Concurrently, UGT84A119, UGT84A49, and J045 were positioned within the UGT84 clade, as anticipated, aligning with group L of the phylogenetic classification (Fig. 2b, Supplementary Table 3).

## Characterization of UGT72/84 enzymes with anticipated natural substrates

Heterologous expression of the chosen UGT candidates in *E. coli* resulted in soluble expression for the proteins UGT84A49, UGT72D1, UGT84A119 and UGT72D7, which were successfully purified utilizing N-terminal 6xHis-tag (Supplementary Fig. 1). While UGTs are classically known as difficult-to-express enzymes accompanied by low yields of isolated

protein[43], the candidates UGT84A119 and UGT72D1 showed remarkably high soluble expression at 45 mg and 20 mg protein per liter of culture medium, respectively (Supplementary Fig. 1). Despite the optimization of the expression host to *E. coli* C43(DE3), tailored for the production of challenging (membrane-associated; toxic) proteins[44], J045 and KZ95 evaded soluble expression and were thus excluded from the study.

The purified enzymes (0.50 mg mL⁻¹) reacted with sinapic acid (1.0 mM) showed distinct family-dependent product patterns (Fig. 3a) for glycosylation from UDP-glucose (2.0 mM). UGT84-members (UGT84A119, UGT84A49) formed a product in a strictly pH-dependent fashion (Fig. 3b), consistent with the previous studies on glucose-ester forming UGTs operating preferably at acidic pH for a suitable reaction equilibrium[26,37]. Reactions of UGT84A119 and UGT84A49 at pH 5.0 yielded a glucosylated product, while no conversion was observed at pH 8.0 (Fig. 3b). As anticipated, preparative scale (13.5 mg) reaction with product isolation confirmed the product identity as sinapic acid glucose ester (Fig. 3a, Supplementary Fig. 2). UGT72-enzymes UGT72D1 and UGT72D7 demonstrated the preference for alkaline pH for glycosylation (Fig. 3c), which is consistent with the pH dependence of the equilibrium constant in favor of glycoside formation at basic pH[45–48]. Product isolation from a preparative (20.5 mg) reaction identified the obtained compound as sinapic acid 4-*O*-glucoside (Fig. 3a, Supplementary Fig. 3). Time course analysis of the reactions performed at suitable pH (5.0 for UGT84A119, UGT84A49; pH 8.0 for UGT72D1, UGT72D7; Fig. 3d) and enzyme concentration

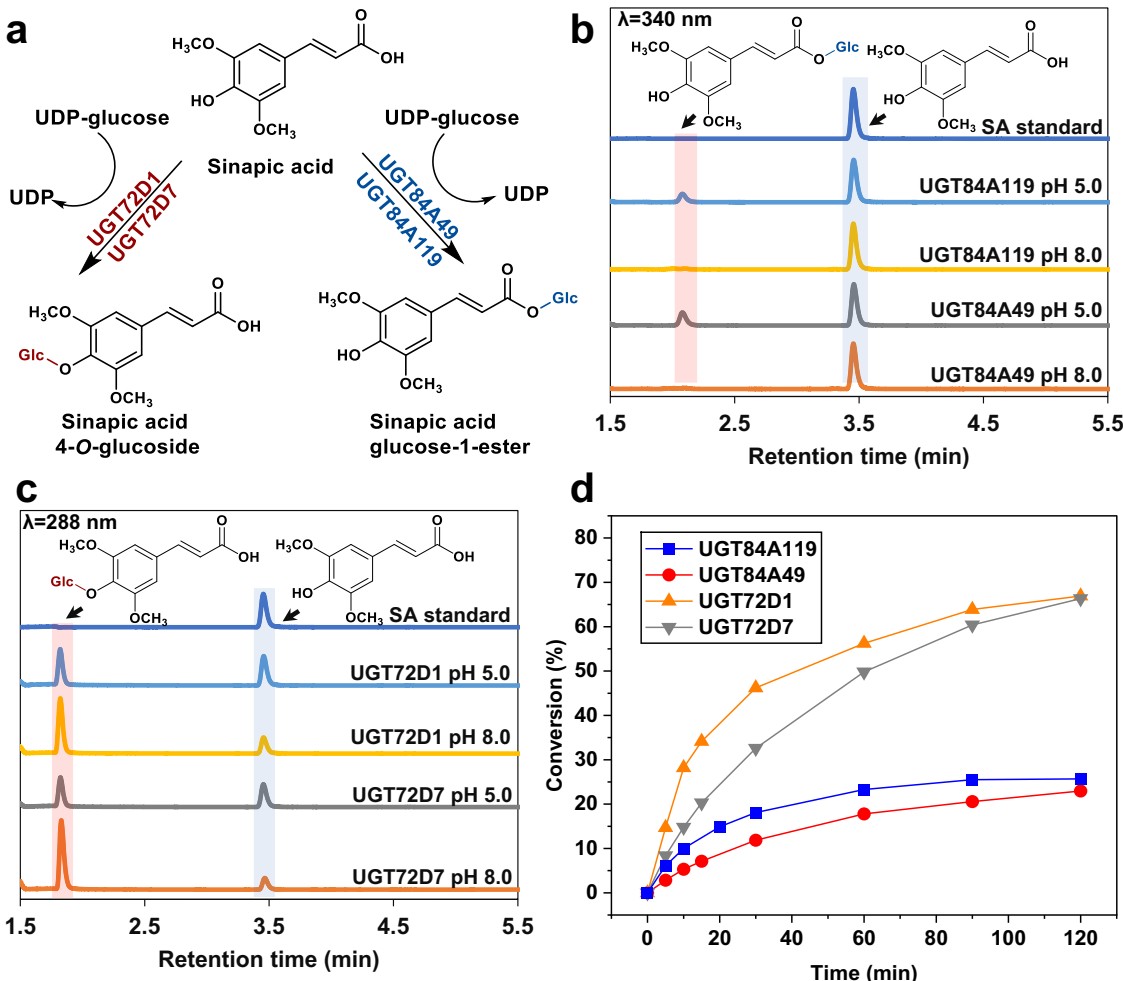

**Fig. 3 | Reaction scheme, overlay of HPLC chromatograms and time courses for UGT reactions with sinapic acid (SA). a** Reaction scheme from sinapic acid to the corresponding 4-*O*-glucoside catalyzed by UGT72 enzymes (UGT72D1, UGT72D7), or to glucose-1-ester catalyzed by UGT84 members (UGT84A119, UGT84A49). **b** Overlay of HPLC chromatograms (24 h time point) from UGT84A119/UGT84A49 reactions with sinapic acid at pH 5.0 and 8.0. **c** Overlaid

HPLC chromatograms (24 h time point) from UGT72D1/UGT72D7 reactions with sinapic acid at pH 5.0 and 8.0. **d** Time courses of product formation in UGT/sinapic acid reactions used for measuring the enzymatic activities. Reactions (100 μL) contained 1.0 mM sinapic acid, 2.0 mM UDP-Glc and 0.0050-0.50 mg mL$^{-1}$ enzyme in 50 mM potassium phosphate buffer (pH 5.0 or 8.0).

allowed the precise measurement of enzymatic activities. UGT84A119 showed a specific activity of 2.0 U mg$^{-1}$ (±2%; $N = 2$) towards sinapic acid, similar to the value reported for the neighboring enzyme UGT84A13 from the same phylogenetic clade (Fig. 2b)[49]. The activity of UGT84A49 (1.0 U mg$^{-1}$ ± 1%; $N = 2$) fell within the same order of magnitude. UGT72D7 catalyzed the glycosylation at the phenolic hydroxyl with specific activity of 0.3 U mg$^{-1}$ (±6%; $N = 2$), comparable to a monolignol glycosyltransferase UGT72E3 (~1.2 U mg$^{-1}$)[19] from the same family (Fig. 2b). Since UGT72D1 exhibited significantly lower activity of 55 mU mg$^{-1}$ (±3%; $N = 2$) toward sinapic acid, coniferyl aldehyde (1.0 mM) was tested as an alternative sugar acceptor from the phenylpropanoid class (Supplementary Fig. 4a). Both UGT72D1 and UGT72D7 showed strong preference for coniferyl aldehyde over sinapic acid, with notable enzymatic activities of 4.9 U mg$^{-1}$ (±10%; $N = 2$) and 1.7 U mg$^{-1}$ (±6%; $N = 2$), respectively (Supplementary Fig. 4b). The product was identified after isolation from a preparative-scale (32.3 mg) reaction as coniferyl aldehyde 4-*O*-glucoside (Supplementary Figs. 5, 4a). While other UGT72 enzymes (e.g., *Nb*UGT72AY1[50], *St*UGT72AY2[50], UGT72E3[23], Fig. 2b) are also capable of producing coniferyl aldehyde 4-*O*-glucoside, the activities have typically been lower (<0.2 U mg$^{-1}$) than those obtained here for UGT72D1 and UGT72D7.

Intriguingly, also the UGT84 enzymes UGT84A119 and UGT84A49 demonstrated modest enzymatic activities of 28 mU mg$^{-1}$ (±6%; $N = 2$) and 5.2 mU mg$^{-1}$ (±4%; $N = 2$) towards coniferyl aldehyde, respectively (Supplementary Fig. 4b). To the best of our knowledge, this observation marks the first instance of such activity within the UGT84 family. The results suggest that the substrate specificity of UGT84 and UGT72 enzymes may be significantly broader than previously understood. All the reactions performed with sinapic acid and coniferyl aldehyde were additionally analyzed for the formation of UDP, which correlated well with the concentration of glycoside product released (Supplementary Figs. 6, 7). Consequently, we concluded that the enzymes characterized in this study did not exhibit hydrolytic side activity when reacted with phenylpropanoid acceptor substrates.

### Exploration of the acceptor scope of UGT72/84 enzymes
To assess the substrate specificity and promiscuity of the discovered UGTs, five characteristic phenolic compounds (Fig. 1b) were chosen to screen for enzymatic activity and resulting product profiles. Specifically, *p*-hydroxyacetophenone (*p*-HAP, Fig. 1b) was selected as a representative acetophenone derivative, given the commercial and medicinal significance of both *p*-HAP and its *O*-glucoside, picein[51,52]. From the group of natural

**Fig. 4 | Overlay of HPLC chromatograms for acceptor substrates and glycoside products in UGT reactions with different polyphenol acceptors. a** UGT reactions with *p*-HAP. **b** UGT reactions with phloretin. **c** UGT reactions with daidzein. All reactions (100 µL) contained 1.0 mM acceptor substrates, 2.0 mM UDP-Glc and 0.50 mg mL⁻¹ enzymes in potassium phosphate buffer (50 mM, pH 8.0).

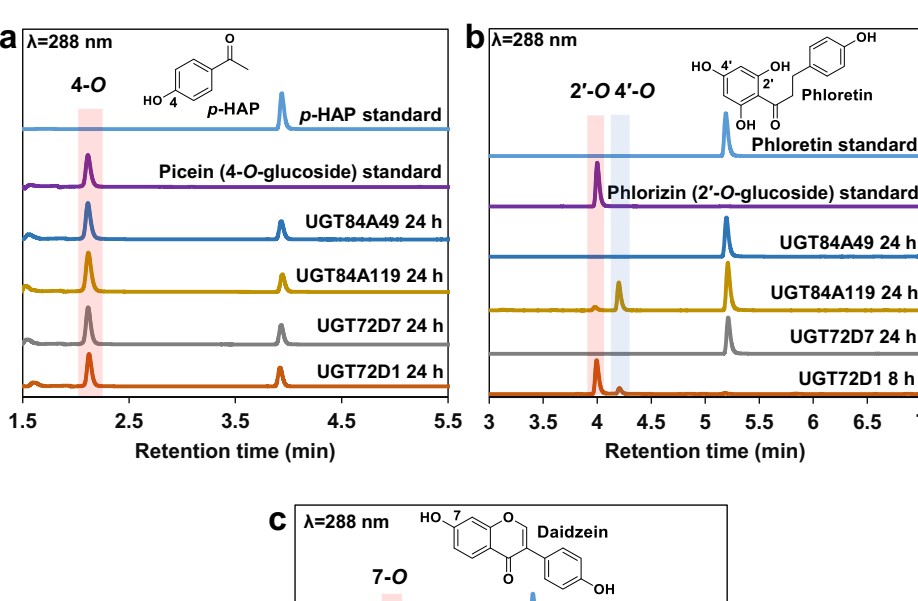

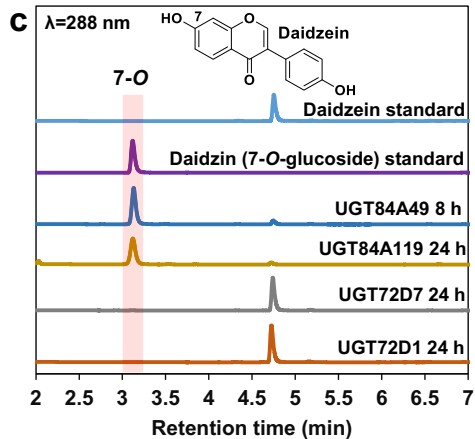

## Table 1 | Activities of UGTs toward phenolic compounds[a]

| UGT | Activities (mU mg⁻¹, $N = 2$) | | | | | |
|---|---|---|---|---|---|---|
| | *p*-HAP | Phloretin | Apigenin | Luteolin | Daidzein | Activity with other sugar donors[c] |
| UGT84A49 | 31.4 ± 0.6 (4-*O*) | /[b] | 34.2 ± 0.8 (7-*O*) <br> 18.9 ± 0.4 (4′-*O*) <br> 7.8 ± 0.8 (7,4′-di-*O*) | 14.7 ± 0.1 (7-*O*) <br> 27.5 ± 0.1 (4′-*O*) <br> 10.2 ± 0.6 (7,4′-di-*O*) | 224 ± 11.7 (7-*O*) | 38.5 ± 0.1 (Daidzein, UDP-Xyl) |
| UGT84A119 | 12.4 ± 0.3 (4-*O*) | 0.4 ± 1.4 × 10⁻³ (2′-*O*) <br> 2.4 ± 1.4 × 10⁻² (4′-*O*) | 64.7 ± 3.0 (7-*O*) <br> 0.3 ± 2.8 × 10⁻² (4′-*O*) <br> 3.9 ± 0.2 (7,4′-di-*O*) | 83.1 ± 0.1 (7-*O*) <br> 5.6 ± 0.3 (3′-*O*) <br> 4.9 ± 0.4 (4′-*O*) <br> 1.3 ± 7.0 × 10⁻² (7,3′-di-*O*) <br> 1.9 ± 0.1 (7,4′-di-*O*) | 171 ± 21.2 (7-*O*) | 9.6 ± 0.6 (Daidzein, UDP-Gal) <br> 6.5 ± 0.4 (Daidzein, UDP-Xyl) |
| UGT72D1 | 171 ± 5.4 (4-*O*) | 34.7 ± 0.4 (2′-*O*) <br> 4.3 ± 0.1 (4′-*O*) | 9.8 ± 0.6 (7-*O*) <br> 4.5 × 10⁻² ± 2.8 × 10⁻³ (7,4′-di-*O*) | 14.2 ± 0.6 (7-*O*) <br> 22.3 ± 0.1 (3′-*O*) <br> 7.5 × 10⁻² ± 3.5 × 10⁻³ (7,3′-di-*O*) | /[b] | 2.9 ± 0.1 (*p*-HAP, UDP-Xyl) |
| UGT72D7 | 2.4 ± 0.2 (4-*O*) | /[b] | /[b] | 0.5 ± 4.5 × 10⁻³ (7-*O*) <br> 7.2 ± 0.8 (4′-*O*) <br> 0.1 ± 3.0 × 10⁻³ (7,4′-di-*O*) | /[b] | /[b] |

[a]Reactions (100 µL) contained 1.0 mM acceptor substrate, 2.0 mM UDP-sugar and 0.50 mg mL⁻¹ enzyme in potassium phosphate buffer (50 mM, pH 8.0), and were carried out at 30 °C without agitation.
[b]"/" stands for no products detected in the reaction.
[c]UDP-Xyl, UDP-Gal and UDP-GlcA were tested. Daidzein was used as acceptor for UGT84A49 and UGT84A119, while *p*-HAP and luteolin were chosen for UGT72D1 and UGT72D7, respectively.

product polyphenols, phloretin was chosen as a dihydrochalcone, daidzein as isoflavone, and both apigenin and luteolin as flavones (Fig. 1b).

Through comparative HPLC chromatogram analysis of the reaction mixtures and standard samples, all four UGTs were found capable of catalyzing the formation of 4-*O*-glucoside of *p*-HAP (Fig. 4a, Supplementary Figs. 8, 9). Thus, *p*-HAP emerges as a potential universal acceptor substrate for both UGT72 and UGT84 family members. The specific activities of the enzymes towards *p*-HAP varied considerably (Table 1, Supplementary

Fig. 9), being at least 30-fold lower than those for phenylpropanoid substrates. While UGT84A49, UGT84A119 and UGT72D7 converted *p*-HAP with activities falling below 32 mU mg⁻¹, UGT72D1 showed a remarkably higher activity of 171 mU mg⁻¹ (Table 1, Supplementary Fig. 9). The results obtained hold considerable interest in the realm of picein biosynthesis. Prior to this study, only one UGT (UGT72E2 from *Arabidopsis thaliana*, Fig. 2b) has been documented to catalyze the formation of this glycoside, with volumetric activity of 18.4 mU mL⁻¹[53]. Notably, the activity measured for

UGT72D1 (85.5 mU mL$^{-1}$) exceeds the value reported for UGT72E2 by 4.6-fold.

Although phloretin is commonly recognized as a substrate by UGTs belonging to the phylogenetic groups L and E (Fig. 2b)[54,55], our findings indicate that only UGT72D1 and UGT84A119 exhibited reactivity towards this particular acceptor (Fig. 4b). UGT84A49 and UGT72D7 proved inactive independent on the enzyme concentration used, while UGT72D1 showed higher activity (40 mU mg$^{-1}$) than UGT84A119 (2.7 mU mg$^{-1}$, Supplementary Fig. 10). Two products were observed in both cases; phlorizin (phloretin 2′-O-glucoside) and herein isolated and characterized phloretin 4′-O-glucoside (Fig. 4b, Supplementary Figs. 11, 12). Notably, the regio-selectivity varied between the enzymes: while the UGT84-member preferred the position 4′-O for glycosylation, the UGT72 enzyme showed clear preference for the position 2′-O (Fig. 4b).

Isoflavones, alongside flavones, constitute another noteworthy group of flavonoids that serve as potential acceptor substrates for UGT72 and UGT84 enzymes. While the UGT72 enzymes (UGT72D1, UGT72D7) were inactive with daidzein, both UGT84 members (UGT84A119, UGT84A49) evidently catalyzed the conversion of daidzein into its 7-O-glucoside daidzin (Fig. 4c). Notably, other members of the UGT84 family, such as PgUGT84A23 from Punica granatum and UGT84F6 from Glycyrrhiza uralensis, have also demonstrated substantial 7-O-glycosylation activity towards isoflavones[26,56]. The specific activities recorded for UGT84A49 (224 mU mg$^{-1}$) and UGT84A119 (171 mU mg$^{-1}$) toward daidzein (Table 1, Supplementary Fig. 13) were in line with the activity of PgUGT84A23 toward an analogous genistein (414 mU mg$^{-1}$)[26]. These findings suggest that members of the UGT84 family may be additionally classified as isoflavone 7-O-glycosyltransferases, corroborating both previous and current studies.

Flavones differ structurally from isoflavones in respect to the positioning of the ring B of the flavonoid skeleton (attached to the C2 instead of C3, Fig. 1b)[57]. Two common flavones, apigenin and luteolin, were chosen for further probing of the acceptor scope of the discovered UGTs. The most interesting reaction characteristics were observed with the UGT84 enzymes, including the potential for di-O-glycosylation. UGT84A49 reacted with apigenin produced two mono-O-glucosides and one di-O-glucoside, confirmed by HPLC-UV (Fig. 5a) and HPLC-UV/MS (Supplementary Fig. 14). Product isolation and subsequent NMR analysis identified the unknown intermediate as apigenin 4′-O-glucoside (Supplementary Figs. 15–17), and the double-glycosylated product as apigenin 7,4′-di-O-glucoside (Supplementary Figs. 18, 19). Similarly, the UGT84A49 reactions with luteolin produced the corresponding 7-O, 4′-O and 7,4′-di-O-glucosides, as identified based on HPLC-UV and NMR analysis (Fig. 5b, Supplementary Figs. 20–22). While the activity of UGT84A49 towards apigenin (61 mU mg$^{-1}$, Supplementary Fig. 23a, b) and luteolin (51 mU mg$^{-1}$, Supplementary Fig. 23c, d) was similar, the striking difference arises from the regio-selectivity: while apigenin was preferably glycosylated at position 7-O (34 mU mg$^{-1}$ towards apigenin 7-O-glucoside, Table 1, Fig. 5c), the reaction with luteolin favored the position 4′-O (28 mU mg$^{-1}$ towards luteolin 4′-O-glucoside, Table 1, Fig. 5d). The formation of the respective di-O-glucosides was considerably slower (7.8 mU mg$^{-1}$ towards apigenin 7,4′-di-O-glucoside; 10.2 mU mg$^{-1}$ towards luteolin 7,4′-di-O-glucoside, Table 1), consistent with previous reports on di-O and di-C-glycosyltransferases capable of catalyzing the serial two-step glycosylation[58,59].

While UGT84A119 exhibited comparable enzymatic activity (68 mU mg$^{-1}$, Supplementary Fig. 24a, b) and similar product pattern (7-O, 4′-O, 7,4′-di-O) with apigenin as observed for UGT84A49, including the preference for the 7-O position (Fig. 5e), its reaction with luteolin was markedly more complex (Fig. 5f). The activity of UGT84A119 (101 mU mg$^{-1}$, Supplementary Fig. 24c, d) was 2-fold higher relative to UGT84A49, and UGT84A119 demonstrated a pronounced affinity for glycosylation at the 7-O position on ring A (83 mU mg$^{-1}$ towards luteolin 7-O-glucoside, Table 1). Beyond the glycosides identified in the UGT84A49 reaction, UGT84A119 produced two additional compounds, characterized as luteolin 3′-O-glucoside and luteolin 7,3′-di-O-glucoside, from a combination of HPLC-MS and NMR analysis (Supplementary Figs. 25, 26). Thorough analysis of the

apigenin and luteolin time courses (Fig. 5e, f) suggests that the formation of di-O-glucosides predominantly initiates from the corresponding 7-O-glucosides. Several other enzymes within the UGT84 family, e.g., UGT84A68 and UGT84A34, have also demonstrated a propensity for synthesizing 7,4′-di-O-glucosides of flavones[58,60].

Contrasting with the UGT84 family members, the UGT72 enzymes exhibited significantly lower activities toward the di-O-glycosylation of flavones (Table 1). UGT72D7 was found to be inactive with apigenin but displayed reactivity with luteolin (8.1 mU mg$^{-1}$, Supplementary Fig. 27), showing a notable preference for the position 4′-O on ring B (7.2 mU mg$^{-1}$ toward luteolin 4′-O-glucoside, Table 1, Supplementary Fig. 27). Interestingly, UGT72D1 was capable of glycosylating both apigenin and luteolin, albeit with lower activities (9.8 mU mg$^{-1}$ and 32 mU mg$^{-1}$, respectively, Supplementary Fig. 28) relative to the UGT84 enzymes. While the glycosylation of apigenin by UGT72D1 predominantly yielded the 7-O-glucoside product, the reaction with luteolin exhibited a substantial preference for the position 3′-O (22 mU mg$^{-1}$ toward luteolin 3′-O-glucoside, Table 1, Supplementary Fig. 28). The di-O-glycosylation activities of UGT72D1 and UGT72D7 towards flavones were low (Table 1), with only trace amounts of di-glycosides detected in the product mixtures (Supplementary Figs. 27b, 28b, e).

These findings are significant in demonstrating that the addition of a single hydroxyl group can dramatically alter the preferred glycosylation site from ring A in apigenin to ring B in luteolin, or induce enzymatic activity into a previously non-accepted substrate (UGT72D7 with apigenin/luteolin). The observed phenomenon may be attributed to the effect of luteolin conformers on the p$K_a$ values of its phenolic hydroxyl groups, which are conventionally ranked by decreasing acidity as follows: 4′-O > 7-O > 3′-O > 5-O[61]. The degree of ionization between the hydroxyl groups at positions 4′ and 7 is remarkably similar (p$K_a$ ~ 7), whereas the hydroxyl group at position 3′ displays significantly lower acidity (p$K_a$ > 10)[61]. Luteolin has been reported to possess three conformers of the B-ring hydroxyl substituents, with the type of conformer having a profound influence on the order of p$K_a$ values[61]. Notably, the p$K_a$ value at position 3′-O is substantially reduced to ~7.5 in one conformer[61]. This suggests that the UGT72 enzyme UGT72D1 likely engages in specific binding interactions to stabilize this particular luteolin conformation, thereby rendering the hydroxyl group at position 3′ accessible for deprotonation by an active site histidine residue (see Supplementary Fig. 29 for a general UGT reaction mechanism). In contrast, UGT72D7 and the UGT84 family members UGT84A119 and UGT84A49 are hypothesized to utilize relatively nonspecific binding interactions, facilitating glycosylation at positions 4′-O and 7-O, achievable from any of the three luteolin configurations. The case of apigenin diverges due to the presence of a single hydroxyl group in ring B. Recent studies have delineated the p$K_a$ values for apigenin's hydroxyl groups as 7.2 (4′-O), 8.7 (7-O), and 11.8 (5-O), determining the position 4′-O as the most acidic[62]. This raises curiosity about the preference for position 7-O by all the enzymes, while explaining the complete absence of glycosylation at 5-O. Although p$K_a$ values may significantly alter upon enzyme binding[63–66], the general order of acidity is anticipated to be preserved in this context. The in-depth time course analysis of the reactions offers valuable insights into the evolving product composition, especially in reactions involving di-O-glycosylation. The UGT84-catalyzed reactions with apigenin culminate in ~70-80% conversion to apigenin 7,4′-di-O-glucoside at the end-point, with apigenin 7-O-glucoside predominating within the first hour of the reaction (Fig. 5c, e). Notably, the UGT84 reactions proceed to complete conversion of apigenin to a mixture of its glycosides (Fig. 5c, e and Supplementary Figs. 23a, 24a), while the UGT72D1 reaction stabilizes at an equilibrium of apigenin (~45%) and apigenin 7-O-glucoside (~55%) at the end-point (Supplementary Fig. 28b). Conversely, for luteolin, all reactions (UGT84A49, UGT84A119, UGT72D7, UGT72D1) advance to complete conversion of the acceptor to its glycosides (Fig. 5d, f and Supplementary Figs. 27a, b and 28d, e), ruling out the potential scenario of enzyme inactivation over prolonged reaction times. Detailed time course analysis further reveals that under the herein applied reaction conditions, the UGT72 and

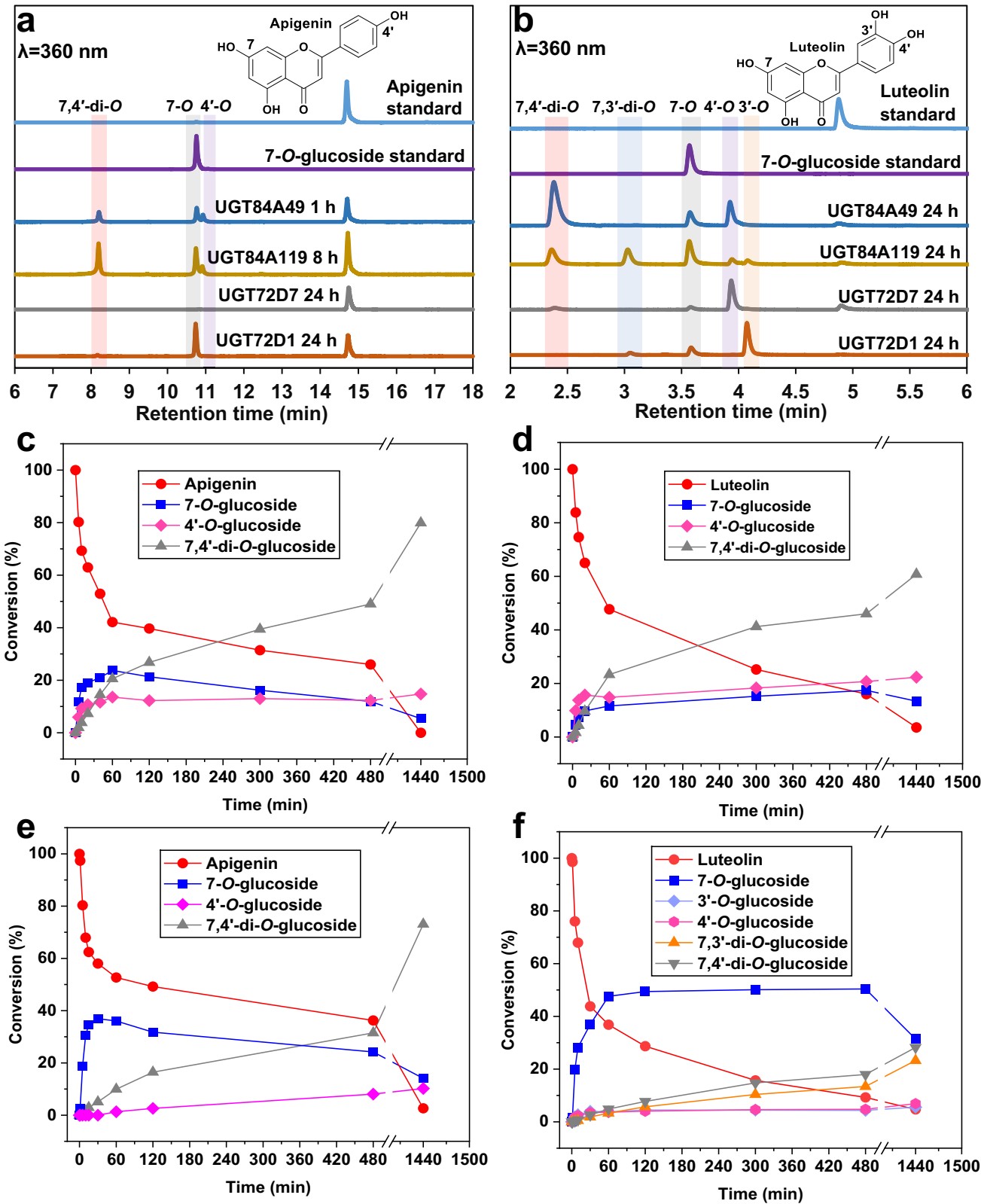

**Fig. 5 | Overlay of HPLC chromatograms and time courses for acceptor substrates and products in UGT reactions toward apigenin and luteolin.** Overlaid HPLC chromatograms for apigenin (**a**) and luteolin (**b**) reactions. Time courses for UGT84A49/apigenin (**c**), UGT84A49/luteolin (**d**), UGT84A119/apigenin (**e**) and UGT84A119/luteolin (**f**) reactions. Reactions (100 μL) contained 1.0 mM apigenin or luteolin, 2.0 mM UDP-Glc and 0.50 mg mL$^{-1}$ enzymes in potassium phosphate buffer (50 mM, pH 8.0).

**Fig. 6 | Active site close-ups and partial multiple sequence alignment showing the conserved donor binding residues in UGTs. a** The donor complex of *Gg*CGT[74] (blue carbons; PDB: 6L5P) with UDP-Glc (gray carbons) overlaid with the AlphaFold-predicted structures of UGT72D1 (pink), UGT84A49 (light green), UGT84A119 (dark cyan) and UGT72D7 (orange). The conserved tetrad Gln-Asp/Glu-Trp-Thr for the binding of glucose C2-OH, C3-OH, C4-OH and C6-OH is seen in *Gg*CGT. **b** Sequence comparison of flavonoid *O*- and *C*-glycosyltransferases indicating the UDP-Glc binding interactions in *Gg*CGT (red, below the alignment) and UGT84A49 (blue, above the alignment). The conserved Thr for stabilization of glucose C6-OH is highlighted in yellow, for UGT84A49 and UGT84A119 the corresponding residue is Ile/Val in the preceding position in the sequence. The black numbering corresponds to the positions in the PSPG box, as show in Fig. 2a.

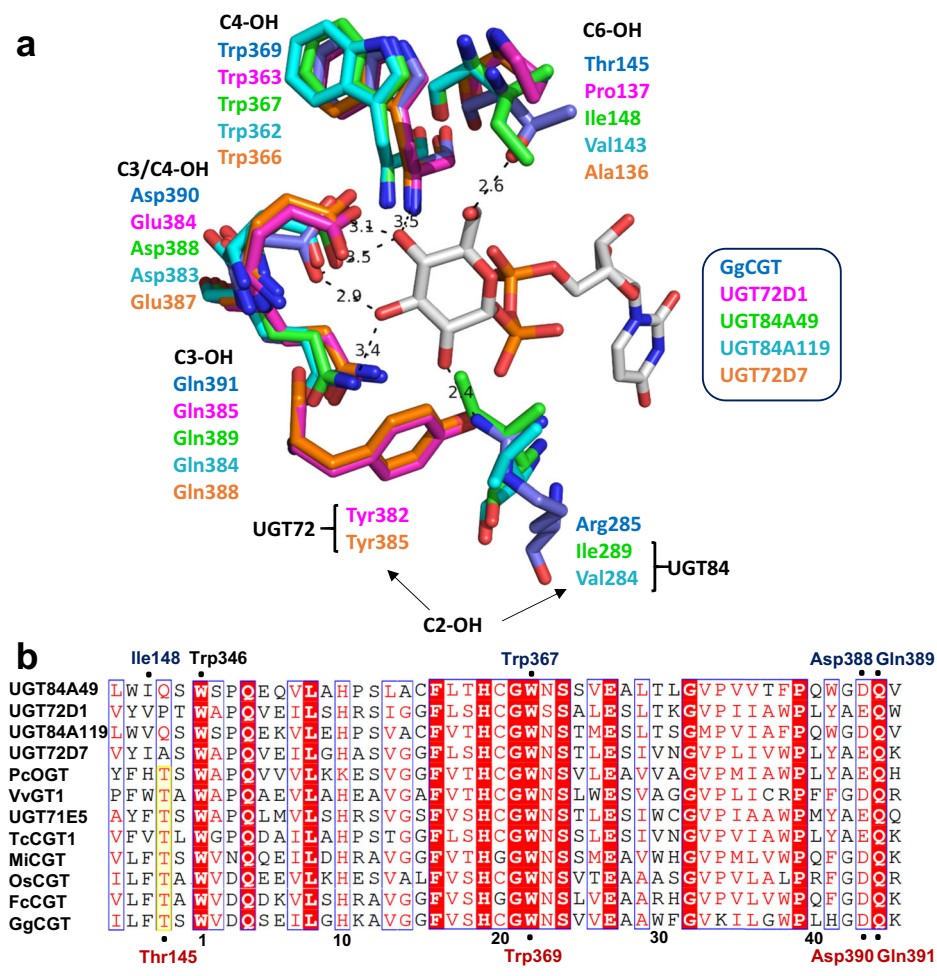

UGT84 reactions with daidzein and *p*-HAP reach an equilibrium consisting of ~90% and ~75% of the corresponding mono-*O*-glucoside products, respectively (Supplementary Figs. 9, 13).

The reactions with apigenin, luteolin, daidzein and phloretin demonstrated high consistency between UDP and glycoside formation, further confirming the accuracy of the reaction analysis. The meticulous analysis of UDP-glucose consumption is critical for two reasons: to track the possible donor substrate hydrolysis, an undesired phenomenon commonly linked to glycosyltransferases[67], and to verify the accuracy of the HPLC-UV analysis for acceptor to glycoside conversion. Given the low water solubility of the selected acceptor substrates and the potential variance in absorption maxima between the acceptors and their glycosides, the reliability of comparing relative peak areas was consistently validated through the ratio of UDP-glucose to UDP. The reactions with *p*-HAP showed significant UDP-glucose hydrolysis (Supplementary Fig. 9), likely due to the small acceptor substrate size causing nonspecific binding and the formation of catalytically inactive complexes, which permit donor hydrolysis but prevent glycosyl transfer. Collectively, the herein obtained results highlight the considerable adaptability of the UGT active sites. Particularly, the UGT84 family exhibits substantial flexibility in acceptor binding, adeptly accommodating both flavones and isoflavones in catalytically productive conformations. It is generally understood that UGTs engage acceptor substrates within hydrophobic cavities, which possibly accounts for the observed multi-site specificity (e.g., 4′-*O* and 7-*O* glycosylation sites for the same substrate) attributed to positional effects[14,68–70].

**Investigation of the donor binding site in UGT72/84 enzymes**

Unlike acceptor binding, which predominantly involves hydrophobic interactions, donor substrates in UGTs typically bind via specific hydrogen bonding interactions, leading to a notably higher specificity for donor compared to acceptor substrates[14,45,69,71]. We evaluated UDP-xylose (UDP-Xyl), UDP-galactose (UDP-Gal), and UDP-glucuronic acid (UDP-GlcA) as alternative sugar donors for UGT84A49, UGT84A119, UGT72D1 and UGT72D7. Our findings revealed that all enzymes, except UGT72D7, utilized UDP-xylose, though the activities were largely reduced compared to UDP-glucose (Table 1, Supplementary Figs. 30, 31). UDP-galactose was accepted by UGT84A119, albeit with a significant decrease in activity (Table 1, Supplementary Fig. 32), while UDP-glucuronic acid was not utilized by any enzyme. UGTs active on UDP-glucose generally feature a conserved Gln-Asp/Glu-Trp-Thr amino acid tetrad, interacting with the hydroxyl groups of the donor glucose (Supplementary Fig. 33)[24,69,72–75]. To elucidate the specific donor binding interactions in the herein characterized enzymes, we employed AlphaFold for quaternary structure prediction. Comparison with the crystal structures of flavonoid *O*-glycosyltransferase *Vv*GT1[73] (Supplementary Fig. 33) and di-*C*-glycosyltransferase *Gg*CGT[74] (Fig. 6a) revealed the absence of the Thr residue, which is commonly associated with stabilizing the glucose C6-OH (Fig. 6a, Supplementary Figs. 33, 34). Notably, UGT84A49 possesses an isoleucine (Ile148) at this position, a residue previously linked to xylosyl- and apiosyltransferases[76,77]. Prior hypotheses suggest that the steric hindrance of isoleucine leads to specificity for UDP-pentose donors, precluding UDP-glucose from fitting into the active site[77]. Our findings indicate that the donor specificity is more intricate than previously understood, as UGT84A49 displays a strong preference for UDP-glucose over UDP-xylose, despite the presence of isoleucine near the glucose C6-OH. Here, we observed that relying solely on the multiple sequence alignment proved insufficient for precise comparison of residues. In the cases of UGT84A49 and UGT84A119, sequence alignment suggested glutamine in the position close to C6-OH (Gln149 and Gln144, respectively), whereas structural predictions identified

the correct candidate as the preceding amino acid in the sequence (Fig. 6b, Supplementary Fig. 34). The universal preference for UDP-glucose among the enzymes indicates that the conserved differences observed in the PSPG motifs of UGT72 and UGT84 members (Fig. 2a, Supplementary Fig. 35) do not play a decisive role in determining donor substrate specificity. This observation is consistent with the findings reported in a recent review[78], which suggest that positions other than 2, 5, 15, and 40 of the PSPG box are considered the most critical in defining the donor scope of UGTs. Additional sequence analysis of UGTs from groups E and L suggested that the Gly/Cys residue at position 15 of the PSPG box in UGT72/UGT84 members may be conserved not only within the subfamily but possibly across the entire group, with Gly prevalent in group E and Cys in group L (Supplementary Fig. 36). Similarly, the presence of Ala at position 2 of the PSPG motif in UGT72 enzymes may be a characteristic common to UGTs from group E (Supplementary Fig. 36). Collectively, these findings are of importance for the sequence-based identification and engineering of UGTs tailored for distinct donor substrate preferences.

## Conclusions

In conclusion, this study has elucidated the glycosylation capabilities and substrate preferences of novel plant UGTs within the UGT72 and UGT84 families, demonstrating their functional versatility and substrate specificity. The identification and characterization of new UGT enzymes contributes to expanding the repertoire of biocatalysts capable of synthesizing natural polyphenol glycosides from UDP-glucose. Through detailed time course and reaction rate analyses combined with product isolations and characterizations, distinct regio-selectivities and specific activities were revealed towards a variety of polyphenolic substrates. Notably, the herein discovered UGT84 enzymes were characterized by their ability to form glucose esters from phenylpropanoid acceptor substrates, and to catalyze the di-O-glycosylation of flavone acceptors. Conversely, the UGT72-members exclusively facilitated the production of phenylpropanoid O-glucosides and lacked the ability of efficient di-O-glycosylation on flavones. The absence of a conserved threonine residue in the AlphaFold-predicted structures of these enzymes, previously associated with donor glucose stabilization, challenges existing paradigms about structure-function relationships within the UGT family. The intricate specificity and selectivity of UGT-mediated reactions with respect to acceptor substrates represent a challenging area that still resists straightforward prediction by means of bioinformatic and structural analyses. This work broadens our understanding of the natural functions and substrate specificity of UGT72 and UGT84 enzymes, offering valuable insights into their potential for the enzymatic synthesis of natural phenolic glycosides. As the complex mechanisms underlying UGT-mediated glycosylation continue to be unraveled, these findings hold the potential to enhance future biotechnological applications, including the design and engineering of UGTs for specific glycosylation reactions.

## Methods

### Bioinformatic analysis

A BLAST search was performed with protein sequences of UGT72E3[19] and UGT84A2[39] as queries, using the online BLAST tool (https://blast.ncbi.nlm.nih.gov/Blast.cgi)[79]. The search was conducted against the NCBI nr and Swiss-Prot database, with a cut-off E-value of $1 \times 10^{-5}$. Multiple sequence alignments were executed using ClustalX software[80], and the visual representation of the aligned sequences was generated using ESPript 3.0[81] or Adobe Acrobat DC software. The phylogenetic tree of characterized plant UGTs (GenBank accession numbers in Supplementary Table 3) was constructed using MEGA 11 software[82], employing the neighbor-joining method based on ClustalW multiple sequence alignment. For enhanced visualization, the tree was modified by the online ornament tool (https://itol.embl.de/)[83]. Structural predictions of the proteins were obtained from AlphaFold DB[84,85].

### Enzyme production

The codon-optimized synthetic genes of candidate UGTs incorporated in pET-28a expression vector were acquired from GenScript (Germany) and transformed into E. coli BL21(DE3) cells. UGTs were expressed in Terrific Broth (TB) medium supplemented with 2% (v/v) glycerol and purified utilizing an N-terminal 6xHis-tag. The size and purity of UGTs were confirmed by SDS-PAGE (Supplementary Fig. 1). Full details of the expression and purification conditions are given in the Supplementary Methods under "Enzyme production".

### Enzyme activity assays

The specific activity of UGTs towards sinapic acid, coniferyl aldehyde, p-HAP, phloretin, apigenin, luteolin and daidzein was determined in an enzymatic assay containing purified UGT enzyme (0.0050-0.50 mg mL$^{-1}$), 1.0 mM acceptor substrate and 2.0 mM UDP-sugar. The initial formation rates for each product were calculated from the corresponding time courses. Full details of the activity assays are provided in the Supplementary Methods under "Enzyme activity assays".

### Preparation and isolation of products

All the preparative reactions were performed using either UGT84A119 or UGT72D1 as a biocatalyst. Sinapic acid 4-O-glucoside and sinapic acid glucose ester were purified using silica column chromatography, coniferyl aldehyde 4-O-glucoside was isolated using TLC. Luteolin 3′-O-glucoside, apigenin 4′-O-glucoside, apigenin 7,4′-di-O-glucoside, luteolin 4′-O-glucoside, and luteolin 7,4′-di-O-glucoside were isolated by preparative HPLC. Phloretin 4′-O-glucoside was separated from the other reaction components using C18-reversed phase column chromatography. Full details of the reactions and product isolations are provided in the Supplementary Methods under "Preparation and isolation of products".

### Analytical methods

**HPLC-UV**. For the separation of UDP-sugars and UDP, a Shimadzu Prominence HPLC-UV system (Shimadzu, Korneuburg, Austria) with Kinetex C18 column (5 μm, 100 Å, 50/150 × 4.6 mm) was employed, using an isocratic method with acetonitrile and tetrabutylammonium bromide (TBAB) buffer (40 mM TBAB, 20 mM $K_2HPO_4/KH_2PO_4$, pH 5.9) as the mobile phase. Detection of UDP-sugars and UDP was achieved by UV at a wavelength of 262 nm. Acceptor substrates and glucoside products were separated using an Agilent 1200 Series HPLC-UV system (Morges, Switzerland) with Kinetex C18 column (3/5 μm, 100 Å, 200/150 × 4.6 mm), using a gradient method of water and acetonitrile (each containing 0.1% (v/v) formic acid) as mobile phase. UV detection for acceptor substrates and glucoside products was performed at wavelength of 288 nm (sinapic acid, coniferyl aldehyde, p-HAP, phloretin, daidzein and corresponding O-glucoside products), 340 nm (sinapic acid and corresponding glucose ester product) or 360 nm (apigenin, luteolin and corresponding O-glucoside products). A summary of the HPLC-UV methods is provided in Supplementary Table 4. The details of the HPLC-UV/MS and preparative HPLC analyses are given in the Supplementary Methods under "Analytical methods–HPLC-UV/MS and Reversed-Phase Preparative HPLC".

**TLC**. Thin-layer chromatography (TLC) was used for the separation of coniferyl aldehyde 4-O-glucoside from its reaction mixture. The samples (10 μL) and reference compounds (2 μL; 40 mM UDP, 40 mM UDP-Glc, 50 mM coniferyl aldehyde) were analyzed on silica gel plates (gel 60 F$_{254}$; Merck, Darmstadt, Germany). The eluent consisting of 1-butanol/acetic acid/deionized water (2:1:1) was used for the separation. Visualization was performed with UV, or DNP-staining consisting of dinitrophenylhydrazine (35% w/v), ethanol (58% v/v), water (24% v/v) and concentrated $H_2SO_4$ (18% v/v).

**NMR**. The identity of the synthesized glycosides was confirmed by NMR. The acquisitions were carried out in deuterium oxide (D$_2$O, 99.96% 2H, Eurisotop, Saint-Aubin Cedex, France), dimethylsulfoxide (DMSO-D6, 99.80% $^2$H, Eurisotop, Saint-Aubin Cedex, France) or mixtures thereof. The spectra of flavonoid glucosides were acquired on a Jeol JNM-ECZL

500 MHz NMR spectrometer with Royal HFX-probe (automatic tuning and matching). Frequency $^1$H: 499.7189 MHz; $^{13}$C: 125.6544 MHz. The Jeol Delta 6.1 software was used for the measurements. The acquisitions of phenylpropanoid glycosides were carried out on a Bruker AVANCE III 300-MHz spectrometer (Bruker, Rheinstetten, Germany) with an auto-sampler. The Bruker Topspin 3.5 software was used for the measurements. The spectra were analyzed using MestReNova 16.0 (Mestrelab Research, S.L.) and JASON (Jeol).

### Reporting summary
Further information on research design is available in the Nature Portfolio Reporting Summary linked to this article.

## Data availability
All relevant data are reported in the manuscript and in the associated Supplementary Information. The file Supplementary Data 1 contains the source data for Figs. 3, 4 and 5. All data are available from the corresponding author upon request. Sequence information on proteins used within this study can be found on the NCBI database (https://www.ncbi.nlm.nih.gov/) using the accession numbers provided in the text.

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

## Acknowledgements

The COMET center acib: Next Generation Bioproduction is funded by BMK, BMAW, SFG, Standortagentur Tirol, Government of Lower Austria und Vienna Business Agency in the framework of COMET—Competence Centers for Excellent Technologies. The COMET-Funding Program is managed by the Austrian Research Promotion Agency FFG.

## Author contributions

A.J.E.B., T.L. and B.N., design of study. T.L., enzymatic reactions and phylogenetic analysis. A.J.E.B., sequence- and structural analysis, HPLC-UV/MS analysis. T.L., isolation of phenylpropanoid glucosides. L.K. and A.J.E.B., isolation of flavonoid glucosides. A.J.E.B. interpreted the NMR spectra acquired by H.W., in collaboration with L.K. R.B., supervision and resources. A.J.E.B., B.N., and T.L. wrote the paper. B.N. funding acquisition.

## Competing interests

The authors declare no competing interests.
