## [Peer Review File · Communications Chemistry]

Reviewers' comments:

Reviewer #1 (Remarks to the Author):

"Discovery, characterization, and comparative analysis of new UGT72 and UGT84 family 1 glycosyltransferases" by Tuo Li and co-workers is a particularly thorough, well-conducted and well-presented characterization of selected UGT enzymes. It is informative, enjoyable to read, the figures are very clear, and as an overall comment I can only congratulate the authors of a very nice piece of work that definitely deserves publication.

It is only showing that I did not find many faults in the authors' work, that my major comment relate to something as trivial as their choice of names for the enzymes they describe. Indeed, while it is clear from their story that there is value in using the devised specific nomenclature system for UGTs, why do the authors decide not to follow it themselves? Using the "correct" names (e.g. UGT84Axx instead of FYZ7) would make the reading much easier, as the reader would always know which subfamily the enzymes belong to. Besides, it only requires an email to the UGT naming committee.

My second comment relate to the hydrolysis comments made by the authors. In l176, the authors refer to figures S6 & 7 to say that there is not much hydrolysis. Figs S6 and 7 concerns themselves with time course of UDP and UDP-Glc and are great to evaluate the quality of their analytics. I can only commend the authors work to follow all substrates and all products of the reactions; it is as rarely done in the field as it is very useful. But to use them to evaluate hydrolysis, one has to compare these results with the ones of Fig 3d. And it is fairly hard to evaluate there if there is hydrolysis or not. If the authors want to make that claim, they can plot on the same figure both products (UDP and glycoside) as either molar or % conversion over time, or both substrate consumption (UDP-Glc and acceptor) as molar amount over time. Alternatively, they could give numbers in supplementary tables, measure the produced amounts of reducing Glc over the reaction, or just decide to tune down the hydrolysis claim.

Figure S9 is much easier to compare with these graphs side by side, and seem to imply considerable amount of hydrolysis for each enzyme except maybe FYZ7 (close to 10% of the UDP-Glc consumption at the end of the reaction). I also do that by generously assuming that something is wrong in the caption, otherwise hydrolysis would even dominate over glycosylation, with each enzyme consuming a higher proportion of 2 mM UDP-Glc than of 1 mM p-HAP to produce the monoglucoside picein. Fig S13d seem also to imply some hydrolysis, with little change in acceptor and glycoside proportions between the 2 and 22h timepoints, while the UDP and UDP-Glc proportion are evolving more.

To sum it up, the authors could moderate or remove their low hydrolysis claims, or should back it up with easier-to-follow data.

After these two quite minor and easy-to-correct comments, I have very few notes.

Miscellaneous comment along the manuscript.

- Abstract, l28, clearly indicate that this is the case for all 4/6 studied enzymes. Also remove "essential" from this line, as this work (and some prior work) clearly shows that it is not essential.

- L 367, I would have liked if the authors extended their comments by having a look with some other members of the families 72 and 84, or even of groups E and L.

- Fig 6, In one of the few UDP structures with bound UDP-Glc, 6SU6.pdb, there is no threonine interacting with the 6 hydroxyl. The closest threonine from OH6 is T146 and is relatively far. I did not look at more crystal structures with bound UDP-Glc but the authors might want to do so, and/or moderate their comments about how "essential" this particular Thr is or how much it relates to donor specificity.

After these few comments, I just want to reiterate my congratulations on a very well made, clear piece of work that deserves publication almost "as is".

Reviewer #2 (Remarks to the Author):

The manuscript by Tuo Li and colleagues reports the thorough characterization of novel plant uridine diphosphate-dependent glycosyltransferases, followed by a study of their atypical donor binding determinants through sequence and structural analysis.

Overall, the work is very comprehensive, well executed, and contains a lot of relevant data. The manuscript is generally well written and the methods are carefully described. The results will be sufficiently interesting to other researchers studying plant glycosyltransferases. I only have a few minor comments.

Specific comments:

1. "Interrogation of the donor binding" – please rephrase
2. An important aspect in the discussion of the sequence/structure analysis is the observation that Ile instead of Thr is present at the structural position of the C6-OH binding residue, because Ile is typically associated with a preference for xylosyl or apiosyl donors. However, in the sequence alignment, that Ile and Thr are not in the exact same position. Is it plausible that the AlphaFold model may be inaccurate? What is the pLDDT in this region of the models?
3. "Consequently, residues 2, 5, 15 and 40 within the PSPG box are not directly involved in substrate binding":
 - a) I am not convinced that this statement is a consequence of the discussed sequence/structural analysis. The fact that they do not play a decisive role in donor specificity does not imply that these residues are not involved in substrate binding at all, which would require some experimental validation.
 - b) It is not easy for the reader to match the mentioned numbers of the PSPG box to the actual residue numbers discussed in the sequence/structure analysis section.
4. I suggest moving the accession numbers for the studied enzymes from the supporting information to the main manuscript to make this information easier to find.

Reviewer #3 (Remarks to the Author):

In this study, Li et al. examine UDP-dependent glycosyltransferases from the families UGT72 and UGT84, chosen because members of these families produce valuable phenolic glycosides and phenolic glucose esters. In total, three plant enzymes from each family were chosen to be investigated, but only two of each expressed as soluble proteins in *E. coli* and were subsequently studied with a panel of phenolic acceptor substrates. The study elucidated substrate specificities and regiospecificities for these UGTs and made use of structure prediction with AlphaFold to examine whether the identity of the residues positioned to interact with a C6-OH of the donor substrate is predictive of donor specificity. The authors findings were contrary to previous suggestions that the presence or absence of a threonine at such a position conferred preference for UDP-glucose or UDP-xylose. The work is solid, though it is not clear to me if the novelty is to the level of Communications Chemistry. I have not read enough of this journal to know its standard and I'd defer to the editor's judgment in this regard.

Some comments:

Figure 1b - The authors could simply put "R" groups where there are purple circles and indicate what R is for the various phenolic compounds.

Figure 2a could include labels indicating the top 4 GTs are UGT84 members and the bottom 4 GTs are UGT72 members

Line 176 - consider revising "Consequently, the enzymes characterized in this study..." to "Consequently, we concluded that the enzymes characterized in this study..."

Is AlphaFold really necessary to determine the presence or absence of a threonine in the key position near C6-OH? Would sequence alignment not suffice?

Response to reviewers

COMMSCHEM-24-0138

Authors: Li T. et al

Title: Discovery, characterization, and comparative analysis of new UGT72 and UGT84 family 1 glycosyltransferases

21-May-2024

General

The authors are grateful for favorable comments and constructive criticism on the manuscript, made by the three Reviewers and the Editor. The revised manuscript considers all of the points raised. Below are our responses to the comments of the Reviewers. The responses are formatted in light blue and are marked with "**Response:**" formatted in bold. The response describes the position of the authors in respect to the point raised and where relevant includes statement about the changes made in the manuscript. We upload a clean copy of the revised manuscript and additionally, a manuscript in which all changes are seen by Track change mode in Word.

Reviewers' comments:

Reviewer #1 (Remarks to the Author):

"Discovery, characterization, and comparative analysis of new UGT72 and UGT84 family 1 glycosyltransferases" by Tuo Li and co-workers is a particularly thorough, well-conducted and well-presented characterization of selected UGT enzymes. It is informative, enjoyable to read, the figures are very clear, and as an overall comment I can only congratulate the authors of a very nice piece of work that definitely deserves publication.

Response: We thank the Reviewer for favorable comment.

It is only showing that I did not find many faults in the authors' work, that my major comment relate to something as trivial as their choice of names for the enzymes they describe. Indeed, while it is clear from their story that there is value in using the devised specific nomenclature system for UGTs, why do the authors decide not to follow it themselves?

Using the "correct" names (e.g. UGT84Axx instead of FYZ7) would make the reading much easier, as the reader would always know which subfamily the enzymes belong to. Besides, it only requires an email to the UGT naming committee.

Response: The authors acquired the correct names from the UGT naming committee: UGT84A119 (FYZ7), UGT84A49 (V6K1), UGT72D1 (ZU72) and UGT72D7 (GVI4). The enzyme names were changed throughout the text and figures in both the main text and Supporting Information.

My second comment relate to the hydrolysis comments made by the authors. In l176, the authors refer to figures S6 & 7 to say that there is not much hydrolysis. Figs S6 and 7 concerns themselves with time course of UDP and UDP-Glc and are great to evaluate the quality of their analytics. I can only commend the authors work to follow all substrates and all products of the reactions; it is as rarely done in the field as it is very useful. But to use them to evaluate hydrolysis, one has to compare these results with the ones of Fig 3d. And it is fairly hard to evaluate there if there is hydrolysis or not. If the authors want to make that claim, they can plot on the same figure both products (UDP and glycoside) as either molar or % conversion over time, or both substrate consumption (UDP-Glc and acceptor) as molar amount over time. Alternatively, they could give numbers in supplementary tables, measure the produced amounts of reducing Glc over the reaction, or just decide to tune down the hydrolysis claim. Figure S9 is much easier to compare with these graphs side by side, and seem to imply considerable amount of hydrolysis for each enzyme except maybe FYZ7 (close to 10% of the UDP-Glc consumption at the end of the reaction). I also do that by generously assuming that something is wrong in the caption, otherwise hydrolysis would even dominate over glycosylation, with each enzyme consuming a higher proportion of 2 mM UDP-Glc than of 1 mM p-HAP to produce the monoglucoside picein. Fig S13d seem also to imply some hydrolysis, with little change in acceptor and glycoside proportions between the 2 and 22h timepoints, while the UDP and UDP-Glc proportion are evolving more.

To sum it up, the authors could moderate or remove their low hydrolysis claims, or should back it up with easier-to-follow data.

Response: The Authors agree that the claims about low hydrolysis appeared misleading. The comment was made based on the reactions with the anticipated natural substrates (coniferyl aldehyde, sinapic acid) where the molar concentration of UDP released matched closely to the concentration of glycoside product formed, within error of max. $\pm 10\%$ (Figure 3d and Figures S4b, S6 & S7). It is to be noted that within this error margin, typically the concentration of UDP released was slightly lower than the concentration of glycoside formed. This indicates that the soluble acceptor substrate concentration in the reaction might have been marginally lower than 1.0 mM in some cases, due to the solubility limit of these compounds. It is true that in the reactions with the selected polyphenols, the p-HAP reactions involved a substantial amount of hydrolysis, as mentioned by the Reviewer in regard to Figure S9. The formation of UDP and glycoside were in good agreement in all the reactions with apigenin, luteolin, phloretin or daidzein as an acceptor substrate, within the error margin of $\pm 10\%$ as with sinapic acid and coniferyl aldehyde. Also in these cases, the error was rather due to the slight inaccuracy of the soluble acceptor substrate concentration (similar to the phenylpropanoid reactions). The reactions with p-HAP (Figure S9) showed hydrolysis dominating over glycosylation (e.g. in the UGT84A49/p-HAP reaction, Fig S9a-b, 1.7mM UDP-Glc consumed vs 0.7 mM glycoside formed, 1.0 mM UDP-Glc lost for hydrolysis). The high hydrolysis rate in the p-HAP reactions might be due to substrate binding in a catalytically inactive conformation, from which UDP-Glc hydrolyses but glycosyl transfer does not take place. It is to be noted that the significant hydrolysis is present exclusively in these reactions with a small acceptor substrate (p-HAP), which can possibly bind in multiple different orientations.

Changes made: As an effort to address these completely valid points of the Reviewer, the authors have both tuned down the low hydrolysis claims in general, and also added a note to the high hydrolysis rate seen in the p-HAP reactions.

After these two quite minor and easy-to-correct comments, I have very few notes.

Miscellaneous comment along the manuscript.

- Abstract, l28, clearly indicate that this is the case for all 4/6 studied enzymes. Also remove “essential” from this line, as this work (and some prior work) clearly shows that it is not essential.

Response: The sentence was modified accordingly (“essential” removed and clearly indicated that the Thr is absent in all four enzymes).

- L 367, I would have liked if the authors extended their comments by having a look with some other members of the families 72 and 84, or even of groups E and L.

Response: The authors extended the comments in regard to PSPG box positions 2, 5, 15 and 40 by comparisons to other family 72 and 84 enzymes, and also group E and L members. A new figure including an additional sequence alignment was placed into the Supporting Information (Figure S35).

- Fig 6, In one of the few UDP structures with bound UDP-Glc, 6SU6.pdb, there is no threonine interacting with the 6 hydroxyl. The closest threonine from OH6 is T146 and is relatively far. I did not look at more crystal structures with bound UDP-Glc but the authors might want to do so, and/or moderate their comments about how “essential” this particular Thr is or how much it relates to donor specificity.

Response: The authors investigated the sequences of more UGTs, and observed that while the Thr is common in UDP-Glc-utilizing enzymes, it is not essential. In addition to the UGT from *Polygonum tinctorium* (PDB: 6SU6), it appears that many UGT72 and UGT84 members do not have Thr in this position (similarly to the enzymes characterized in this study). For example, the enzymes UGT84A3, UGT84B1, UGT84A2 and UGT72E3 all have either valine or isoleucine in this position. The authors have tuned down the statements about the threonine being “essential”, and instead write that it is “commonly found” in UDP-glucose-utilizing glycosyltransferases.

After these few comments, I just want to reiterate my congratulations on a very well made, clear piece of work that deserves publication almost “as is”.

Response: Again, we are happy to receive positive feedback and are grateful for favorable comment.

Reviewer #2 (Remarks to the Author):

The manuscript by Tuo Li and colleagues reports the thorough characterization of novel plant uridine diphosphate-dependent glycosyltransferases, followed by a study of their atypical donor binding determinants through sequence and structural analysis.

Overall, the work is very comprehensive, well executed, and contains a lot of relevant data. The manuscript is generally well written and the methods are carefully described. The results will be sufficiently interesting to other researchers studying plant glycosyltransferases. I only have a few minor comments.

Response: We thank the Reviewer for favorable comment/feedback on our manuscript.

Specific comments:

1. “Interrogation of the donor binding” – please rephrase

Response: The title was rephrased to “Investigation of the donor binding site” instead of “Interrogation of the donor binding”.

2. An important aspect in the discussion of the sequence/structure analysis is the observation

that Ile instead of Thr is present at the structural position of the C6-OH binding residue, because Ile is typically associated with a preference for xylosyl or apiosyl donors. However, in the sequence alignment, that Ile and Thr are not in the exact same position. Is it plausible that the AlphaFold model may be inaccurate? What is the pLDDT in this region of the models?

Response: The pLDDT in this region of the models is very high (94.56–97.27) in all the cases. A new figure showing the corresponding pLDDT scores was included in the Supporting Information (Figure S33). While being an invaluable tool in the analysis of proteins, the Authors have made the experience that sequence alignments might sometimes fail (as in the case shown in this manuscript). Another example of this relates to the commonly observed His-Asp catalytic dyad in UGTs – simple sequence alignment does not always recognize the conserved Asp in the exact same position in all the sequences, while the structural alignments reveal the presence of the conserved Asp. A recent study (Curr. Opin. Struct. Biol., 2023, 102539, <https://doi.org/10.1016/j.sbi.2023.102539>) has compared the accuracy of sequence vs structural (AlphaFold) alignments, highlighting the accuracy of AlphaFold.

3. “Consequently, residues 2, 5, 15 and 40 within the PSPG box are not directly involved in substrate binding”:

a) I am not convinced that this statement is a consequence of the discussed sequence/structural analysis. The fact that they do not play a decisive role in donor specificity does not imply that these residues are not involved in substrate binding at all, which would require some experimental validation.

Response: The authors agree that a claim about the involvement of the corresponding residues in donor binding cannot be made. While they do not play a role in donor specificity, they might still contribute to binding. The sentence was removed.

b) It is not easy for the reader to match the mentioned numbers of the PSPG box to the actual residue numbers discussed in the sequence/structure analysis section.

Response: To make it easier to match the actual residue numbers and the PSPG box numbers, the following action was taken: 1) PSPG box numbering was added into Figure 6b, and 2) A new figure was included in the Supporting Information (Figure S34) showing the actual residues in the PSPG box positions 2, 5, 15 and 40.

4. I suggest moving the accession numbers for the studied enzymes from the supporting information to the main manuscript to make this information easier to find.

Response: The accession numbers were moved from the Supporting Information to the main text section “Discovery and phylogenetic analysis of putative UGT72 and UGT84 enzymes”.

Reviewer #3 (Remarks to the Author):

In this study, Li et al. examine UDP-dependent glycosyltransferases from the families UGT72 and UGT84, chosen because members of these families produce valuable phenolic glycosides and phenolic glucose esters. In total, three plant enzymes from each family were chosen to be investigated, but only two of each expressed as soluble proteins in *E. coli* and were subsequently studied with a panel of phenolic acceptor substrates. The study elucidated substrate specificities and regiospecificities for these UGTs and made use of structure prediction with AlphaFold to examine whether the identity of the residues positioned to interact with a C6-OH of the donor substrate is predictive of donor specificity. The authors findings were contrary to previous suggestions that the presence or absence of a threonine at such a position conferred preference for UDP-glucose or UDP-xylose. The work is solid, though it is not clear to me if the novelty is to the level of Communications Chemistry. I have not read enough of this journal to know its standard and I'd defer to the editor's judgment in this regard.

Response: We thank the Reviewer for favorable comment/feedback on our manuscript. As far as novelty within the broader field of research is concerned, we would like to refer to our Introduction. In this section of the manuscript, we define the place of the study and also summarize briefly the advance made. The Abstract and the Conclusions summarize these advances in a concise manner. While there have been many studies of plant glycosyltransferases, the specificity of these important enzymes is not well understood. Here, we posit, the current study makes a significant contribution. Lastly, Communications Chemistry involves a two-tier process where the Editors decide first whether a manuscript is suitable for external review. Only after this first decision, the manuscript is sent to Reviewers.

Some comments:

Figure 1b - The authors could simply put "R" groups where there are purple circles and indicate what R is for the various phenolic compounds.

Response: The purple circles were removed and the "R" groups added into Figure 1b.

Figure 2a could include labels indicating the top 4 GTs are UGT84 members and the bottom 4 GTs are UGT72 members

Response: Labels were added to indicate the UGT72 and UGT84 family members in Figure 2a.

Line 176 - consider revising "Consequently, the enzymes characterized in this study.." to "Consequently, we concluded that the enzymes characterized in this study.."

Response: The sentence was revised as suggested.

Is AlphaFold really necessary to determine the presence or absence of a threonine in the key position near C6-OH? Would sequence alignment not suffice?

Response: For the presence/absence of the respective threonine (in the enzymes shown in this manuscript), sequence alignment would be sufficient. However, for identifying the actual residue in the corresponding position (instead of threonine), AlphaFold becomes necessary. As discussed in the manuscript, the sequence alignment did not correctly identify the respective residue in the UGT84A49 and UGT84A119 enzymes (glutamine was proposed for both based on the sequence, while the structural prediction identified the residues as isoleucine and valine, respectively).

REVIEWERS' COMMENTS:

Reviewer #1 (Remarks to the Author):

The manuscript was very close to be publishable "as is" before, it certainly is now. I do feel that the points raised in the previous round of review have been satisfactorily addressed.

Reviewer #2 (Remarks to the Author):

The authors have adequately responded to my minor comments. I recommend publication in Communications Chemistry.

COMMSCHEM-24-0138A

Discovery, characterization, and comparative analysis of new UGT72 and UGT84 family glycosyltransferases

Response to reviewers

Our responses are given below in light blue font.

REVIEWERS' COMMENTS:

Reviewer #1 (Remarks to the Author):

The manuscript was very close to be publishable "as is" before, it certainly is now. I do feel that the points raised in the previous round of review have been satisfactorily addressed.

Response: We are grateful for the positive response. No changes were necessary.

Reviewer #2 (Remarks to the Author):

The authors have adequately responded to my minor comments. I recommend publication in Communications Chemistry.

Response: We are grateful for the positive response. No changes were necessary.